# The Impact of Socioeconomic Status and Comorbidities on Non-Melanoma Skin Cancer Recurrence After Image-Guided Superficial Radiation Therapy

**DOI:** 10.3390/cancers16234037

**Published:** 2024-12-01

**Authors:** Liqiao Ma, Michael Digby, Kevin Wright, Marguerite A. Germain, Erin M. McClure, Francisca Kartono, Syed Rahman, Scott D. Friedman, Candace Osborne, Alpesh Desai

**Affiliations:** 1Tru-Skin Dermatology, Austin, TX 78731, USA; 2Renew Family Dermatology, Fort Payne, AL 35968, USA; 3The Clinic for Dermatology & Wellness, Medford, OR 97504, USA; 4Germain Dermatology, Mt Pleasant, SC 29464, USA; 5University Hospitals Geauga Medical Center, Chardon, OH 44024, USA; 6MI Skin Center, Northville, MI 48167, USA; 7Corewell Health, Dermatology Residency, Farmington Hills, MI 48336, USA; 8Trinity Health Ann Arbor Hospital, Ypsilanti, MI 48197, USA; 9Michigan State University College of Osteopathic Medicine, East Lansing, MI 48824, USA; 10Corewell Health Trenton Hospital, Trenton, MI 48183, USA; 11Henry Ford Wyandotte Hospital, Wyandotte, MI 48192, USA; 12Trinity Health, Pontiac, MI 48341, USA; 13McLaren Oakland, Pontiac, MI 48342, USA; 14Tru-Skin Dermatology, Hallettsville, TX 77964, USA; 15Orlando College of Osteopathic Medicine, Winter Garden, FL 34787, USA; 16Heights Dermatology, Houston, TX 77008, USA

**Keywords:** non-melanoma skin cancer, image-guided superficial radiation therapy, freedom from recurrence, deprivation, comorbidity, socioeconomic status, zip code, health disparities, basal cell carcinoma, squamous cell carcinoma, rural, rural–urban continuum code, area deprivation index, Charlson comorbidity index

## Abstract

Image-guided superficial radiation therapy (IGSRT) is an emerging treatment option for non-melanoma skin cancers (NMSCs). The aim of this retrospective cohort study was to assess if there are relationships between patient comorbidities or socioeconomic status (SES) and outcomes from IGSRT treatment for their NMSCs. Data from 19,988 NMSCs revealed no difference in freedom from recurrence in less vs. more deprived neighborhoods (as a measurement of SES) or in patients without comorbidities vs. with many and/or severe comorbidities. This supports the use of IGSRT regardless of SES or comorbidities.

## 1. Introduction

Non-melanoma skin cancers (NMSC) consist primarily of basal cell carcinomas (BCCs) and squamous cell carcinomas (SCCs) and are the most common form of cancer in the United States [1,2]. More than 4 million NMSCs are diagnosed annually in the United States [3], and the incidence and cost of care for NMSCs are increasing worldwide [1,2,4]. NMSC is typically curable with surgery and/or radiation therapy and has a favorable prognosis [5]. However, prognosis is optimized when skin cancers are diagnosed and treated early. This minimizes the risk of local and distant spread of disease. Detection techniques include dermoscopy, confocal microscopy, and computer-aided diagnosis, with the current gold standard being biopsy [6,7]. Unfortunately, even with timely treatment, NMSCs are associated with an increased risk of other malignancies [8,9,10]. SCC has also been correlated with increased all-cause mortality, and SCC and BCC have both been connected to worse survival from second primary malignancies [5].

According to current National Comprehensive Cancer Network guidelines, localized low- and high-risk BCCs and SCCs should be treated with surgical excision, Mohs micrographic surgery (MMS), or radiation therapy. MMS is a precise and tissue-sparing technique for removing skin cancer, offering microscopic control of the entire tumor margin while conserving as much healthy tissue as possible [11]. MMS is preferred over superficial radiation therapy (SRT, a low-energy kilovoltage photon treatment confined to the skin), given its superior outcomes, but SRT remains an option for patients who are poor surgical candidates or prefer a nonsurgical approach [12,13].

Image-guided SRT (IGSRT) was cleared by the United States Food and Drug Administration in 2015 for use in NMSC. IGSRT employs integrated, high-resolution dermal ultrasound technology to improve lesion visualization and enable more precise radiation targeting compared with SRT. With a median dose of 5188 cGy delivered over 20 fractions, IGSRT has shown a 99% local tumor control with a median follow-up of 63 weeks in 2917 NMSC lesions [14]. A retrospective cohort study indicated that IGSRT is a clinically equivalent alternative to MMS for early-stage NMSCs [15].

Social determinants of health, such as low socioeconomic status (SES) and rural residence, have established adverse effects on cancer screening, diagnosis, treatment, and outcomes [16,17,18,19,20,21]. SES can be analyzed at the neighborhood level. Neighborhood-level factors include median annual income, median housing value, unemployment percentage, and percentage of adults without high school completion [22]. Such factors can be quantified via the Area Deprivation Index (ADI) [23]. Higher ADI scores are indicative of worse neighborhood deprivation and have been correlated with increased prevalence of preterm birth and low birth weight [24], an increase in conditions like cardiovascular disease, obesity, and diabetes [25], and worsened progression-free and cancer-specific survival [26].

The degree of residence rurality can be captured by Rural–Urban Continuum Codes (RUCCs) [27]. The data on the impact of rural residence on patterns of care for cancer, stage at diagnosis, and survival are mixed [28]. However, one consistent finding was that patients with breast, endometrial, or prostate cancer were less likely to receive curative-intent radiation [28]. This may be due to rural populations not living near radiation centers, compounded by the fact that radiation treatment typically occurs several times a week for up to 6 weeks [28]. This has not been well-studied in NMSC populations, but there is likely a need for improved access to radiation therapy for NMSC in rural regions. This need is compounded by the fact that Mohs micrographic surgeons are scarce in rural America. A study on 2240 practicing Mohs surgeons who accepted Medicare patients found that the vast majority (*n* = 2118 (94.6%)) practiced in metropolitan areas, whereas only 111 (5%) were in nonmetropolitan areas, and a mere 11 surgeons (0.4%) were operating in rural areas [26].

Comorbidities are also important predictors of cancer outcomes [16,29]. One example is their negative impact on treatment response and options, as severe comorbidities can increase treatment toxicity, limit the survival benefits anticipated with appropriate treatment, and prevent the application of more aggressive treatment options [29]. Comorbidities and SES interrelate and affect patient diagnosis, treatment, and outcomes in complex ways. For instance, living in more deprived neighborhoods is itself a risk factor for a variety of comorbidities [30,31,32]. This highlights the importance of studies evaluating how both SES and comorbidities impact health outcomes.

Given the promising results found so far regarding IGSRT in the treatment of early-stage low- and high-risk NMSC, there is a need for additional research regarding the potential influence of patient and disease characteristics on its efficacy and safety. The impact of SES and comorbidities on the outcomes of cancers treated with non-IGSRT therapies (surgery, chemotherapy, immunotherapy, etc.) has been extensively studied [33,34,35,36]. However, there are no published studies reporting data on IGSRT outcomes with respect to SES or comorbidity burden. Therefore, this study sought to fill an important knowledge gap. Specifically, the objective of this large retrospective cohort study was to determine the effects of neighborhood-level socioeconomic deprivation and comorbidities on freedom from recurrence in patients with ISGRT-treated NMSC.

## 2. Materials and Methods

### 2.1. IGSRT Treatment Methodology and Energy/Dose Selection Process

The IGSRT methodology has been described in detail [14,37]. It follows a general guideline (the Ladd–Yu protocol) [14] for treatment dose, energy, fractionation, and therapeutic biological effect and the Dermatology Association of Radiation Therapy (DART) Appropriate Use Criteria [38,39]. High-resolution dermal ultrasound (HRDUS) is performed during the initial simulation for treatment planning. Daily pre-treatment consists of HRDUS prior to “beam-on” to assess/confirm tumor configuration/location and detect tumor changes, which may indicate the need for a prescription change with adaptive radiation treatment planning. A standardized protocol with a total of ~20 fractions using single energy or a sequential combination of 50 kVp, 70 kVp, or 100 kVp energy X-ray treatment is generally delivered 2–4 times per week with HRDUS. HRDUS is also performed at follow-up visits after treatment course completion to evaluate response.

### 2.2. Tumor Configuration and Depth Determination

HRDUS uses a non-invasive 20–22 MHz ultrasound with a Doppler component probe that is intrinsic to the IGSRT unit (Sensus SRT-100 Vision), which allows imaging of 0–10 mm into the skin structure, which captures the epithelium, papillary dermis, and sometimes down to the reticular dermis depending on anatomic location and skin thickness. This high-resolution/high-frequency ultrasound allows for the clear visualization of the disrupting tumor, which is hypoechoic, typically without Doppler color speckles, and allows for precise tumor size measurements (including depth, width, and breadth), which reduces the risk of anatomical miss and radiation misadministration.

### 2.3. Data Collection

Data collection followed a similar process as that described in previous studies [14,37]. IGSRT treatment records of over 11,000 patients with 19,988 NMSC lesions treated at multiple institutions across the continental United States between 2016 and 2023 were retrospectively gathered. Exclusion criteria include cases missing pertinent documentation (i.e., treatment chart, simulation statistics like time, dose, and fractionation) or stage 3 tumors with deep invasion, cortical erosion, or perineural invasion. Patient characteristics, treatment parameters (treatment site, energy, dose), and NMSC lesion information (recurrence) were extracted manually and accessed electronically from written and electronic medical records (EMRs) for all institutions. Additional data from the EMRs, including race, ethnicity, past medical history, past surgical history, medications, follow-up dates, zip codes, comorbidities, and mortality status/expiratory dates, were “data scraped” with algorithmic programming conducted by a company that offered data scraping services, formerly known as Sympto Health, Inc., currently merged into the healthcare navigation platform, Rely Health.

### 2.4. Area Deprivation Index

SES was measured using the Area Deprivation Index (ADI) [24], a measure of socioeconomic neighborhood deprivation that was developed by the Health Resources and Services Administration using principal components analysis. Scores range from 0 to 100, with higher scores indicating more neighborhood-level socioeconomic deprivation. ADI scores were derived for each participant by comparing their home zip code to a collection of publicly available ADI datasets [40].

### 2.5. Rural–Urban Continuum Codes

RUCCs are a measure of the population size of metropolitan (metro) areas within metro counties or of nonmetropolitan populations, their adjacency to a metro area, and their degree of urbanization [27]. Table 1 delineates RUCC criteria in more detail. RUCCs for this cohort were determined by mapping patient zip codes to county Federal Information Processing Standard (FIPS) codes via the published 2023 FIPS codes on the United States Census Bureau’s webpage [41]. Then, the FIPS were cross-referenced with the Neighborhood Atlas database to determine corresponding RUCCs [42].

### 2.6. Charlson Comorbidity Index

The Charlson Comorbidity Index (CCI) is a commonly used comorbidity assessment tool in studies of the NMSC population and has been extensively validated in other cancer populations [43]. The CCI includes 19 conditions: myocardial infarction, congestive heart failure, peripheral vascular disease, cerebrovascular disease, dementia, chronic pulmonary disease, connective tissue disease, peptic ulcer disease, liver disease, diabetes, hemiplegia, renal disease, any tumor, leukemia, lymphoma, and AIDS [43]. Conditions are weighted differently depending on severity, ranging from 1 to 6. The total CCI score is a sum of these weights. Patients’ health records were examined for CCI-relevant comorbidities, which were then calculated into CCI scores. Higher scores represent an increased mortality risk and the presence of more severe comorbid conditions [44].

### 2.7. Statistical Analysis

Detailed logs of NMSC recurrences were maintained through dermatology practices. These recurrence logs were used to quantify recurrence events. Freedom from recurrence was estimated using the Kaplan–Meier method. ADI and CCI scores were calculated. ADI scores were split into above 50 (high neighborhood deprivation) or equal to/below 50 (low neighborhood deprivation). CCI scores were grouped by like scores ranging from 0 to 4 in this cohort. Groups were compared with respect to freedom from recurrence using the log-rank test.

### 2.8. Ethics

The ethics committee/Institutional Review Board (IRB) of WIRB-Copernicus Group (WCG™) waived ethical approval for this work. The dataset was de-identified prior to analysis, and all data personnel adhered to the Health Insurance Portability and Accountability Act (HIPAA) and ethical standards to protect patient information.

## 3. Results

### 3.1. Patient and Disease Characteristics

Patient and disease characteristics are summarized in Table 2. A total of 19,988 lesions were included. Patients were predominately male (61.7%), and most patients (84.2%) were at least 65 years of age (median age 74.9 years). The majority of lesions (63.7%) were located in the head or neck, a known high-risk location for BCC and SCC [45]. Most lesions were categorized as stage 0 (i.e., SCCIS; 23.4%) or stage 1 (65.7%); 49.5% of lesions were BCC, 26.4% were SCC, 23.2% were squamous cell carcinoma in situ (SCCIS), and there were ≥2 NMSC types in 1% of lesions. Table 3 quantifies the comorbidities in this cohort that contributed to the CCI score. The most common comorbidities were diabetes (17.8%), non-NMSC solid tumor malignancy (7.7%), and stroke or transient ischemic attack (4.6%). The least common comorbidities are omitted from Table 3 and are the following: congestive heart failure (0.8%), chronic pulmonary disease (0.5%), dementia (0.4%), chronic kidney disease (0.3%), peripheral vascular disease (0.1%), connective tissue disease (0%), hemiplegia paraplegia (0%), AIDS (0%), and peptic ulcers (0%). Most NMSCs were found in patients with a CCI of 3 or 4 (28.9% and 28.8%, respectively). The median ADI for the cohort was 50.0. Over half of this cohort (53.7%) reside in RUCC 1 regions, and a fifth reside in RUCC 2 regions (20.7%).

### 3.2. Freedom from Recurrence Rates by Neighborhood Deprivation

Freedom from recurrence rates by neighborhood deprivation (ADI score) at 2-, 4-, and 6-year follow-up can be seen in Table 4, and the association between neighborhood deprivation and freedom from recurrence across the 6 years is illustrated in Figure 1. In NMSC lesions diagnosed in patients with an ADI percentile ≤50, freedom from recurrence was reported in 99.64% at 2 years and 99.47% at 4 and 6 years. In lesions diagnosed in patients with an ADI percentile >50, freedom from recurrence was reported in 99.76% at 2 years and 98.61% at 4 and 6 years. These differences were not statistically significantly different (*p* = 0.2).

### 3.3. Freedom from Recurrence Rates by Comorbidity

Freedom from recurrence rates by comorbidities (CCI score) at 2-, 4-, and 6-year follow-ups are summarized in Table 4, and the association between comorbidities and freedom from recurrence is illustrated in Figure 2 and Figure 3. Specifically, Figure 2 demonstrates the freedom from recurrence data for all CCI scores in this cohort (0–7+). However, due to the grouping of scores 0–6 far from CCI = 7, Figure 3 magnifies the data of CCI scores 0–6+ to better visualize these results. Freedom from recurrence ranged from 99.67% at 2, 4, and 6 years for a CCI of 0 to 99.27% at 2, 4, and 6 years for a CCI of ≥7. These differences were not statistically significant (*p* = 0.9).

## 4. Discussion

The objective of this large retrospective cohort study was to determine the effects of neighborhood-level socioeconomic deprivation and patient-level comorbidities on freedom from recurrence in patients with IGSRT-treated NMSC. In summary, there was no significant association found between neighborhood deprivation or comorbidity burden and freedom from recurrence of NMSC at 2-, 4-, or 6-years following treatment with IGSR. This suggests that IGSRT may be a viable treatment option for patients regardless of their socioeconomic context or severity/number of comorbidities.

The distribution of this cohort’s CCI scores is skewed toward higher scores compared with what is found in the literature. For example, one study on 181,764 hospital workers in the northeastern United States found that 70.2% (*n* = 88,446) of them had a CCI of 0 or 1, 20.6% (*n* = 25,907) had a CCI of 2 or 3, 4.3% had a CCI of 4, and 5% (*n* = 6196) had a CCI of ≥5, whereas this study’s cohort had a CCI of 0 in 1.8% of patients, 1 in 5.6%, 2 in 18%, 3 in 28.9%, 4 in 28.8%, 5 in 10.2%, and ≥6 in 6.5% [46]. An Australian study on over 400,000 hospital admissions found that approximately 76% of them had a CCI of 0, 12% a CCI of 1, 6% a CCI of 2, 2% a CCI of 3, 1% a CCI of 4, 2% a CCI of 5, and 0.7% a CCI of ≥6 [47]. A higher proportion of higher CCI scores in this study compared with the literature may be due to the older patient population present in this study. This is because older age increases the CCI score, and NMSCs are more common in older adults.

Research into treatment efficacy and safety in patients with NMSC in relation to comorbidities is limited. In one small retrospective cohort study of 198 patients with NMSC (median age 78 years), 85.8% of patients presented with at least one comorbidity; hypertension was the most common (68.7%), followed by hyperlipidemia (20.7%) and diabetes (19.7%) [48]. Collectively, these 198 patients had 537 MMS and defect repair operations, and while no deaths occurred, hypertensive episodes during operation occurred in 45.5%, and temporary cardiovascular abnormalities (e.g., tachycardia, bradycardia) occurred in 23.1%. This is partially consistent with our data set since diabetes was found in 17.8% of the cohort. However, hypertension and hyperlipidemia were not specifically studied since the CCI does not include these comorbidities and instead captures more severe, potentially downstream comorbidities such as myocardial infarction, stroke, and peripheral vascular disease. Additionally, while cardiovascular complications can be seen with radiation near the heart (like with breast cancer treatment) or major vessels [49], the acute side effects of SRT for skin cancer are typically cutaneous [50].

Multiple studies have found that the association between neighborhood-level deprivation (lower SES) and poor cancer outcomes remains even after adjustment for other factors, including age, stage, treatment, grade, diagnosis year, race, and rural residence [51,52,53]. Fortunately, this study found no difference in freedom from recurrence of NMSCs following IGSRT when analyzed by neighborhood-level deprivation. While studies on NMSC treatment outcomes and SES associations are lacking, thereby preventing direct comparisons of this study to the existing literature, there is a study on over 39,000 patients treated for melanoma that found a statistically significant decrease in 10-year overall survival when comparing SES quintiles [54]. The highest SES quintile had a 69% survival, with the lowest having a 50% survival. Interestingly, this study found that most patients (99%) received surgery across SES groups, and there were no statistically significant associations between receipt of RT and SES, supporting surgery and RT as relatively accessible treatments. This supports that it is possible for worse outcomes in lower SES populations, despite similar treatment for early-stage cancers. There was an association between high SES and chemotherapy and immunotherapy (*p* = 0.0002 and 0.0092, respectively), which may be explained by such therapies being more expensive and more challenging to access as they are typically found at cancer centers. Systemic treatments are reserved for advanced disease; therefore, it would be interesting for a future study to evaluate if the worse survival in low SES groups is predominantly present in late-stage cancers, or if patients with early-stage cancers also experience worse outcomes relative to their high SES peers.

Rural residence was identified as a potential risk factor for health disparities in a 2003 report by the Institute of Medicine [20]. This report stated that rural hospitals are characterized by lower-quality clinical decision-making, technical diagnostic and therapeutic processes, and monitoring processes than those in urban teaching hospitals [20]. With respect to treatment, women with breast cancer living further from radiation therapy centers have been shown to be more likely to receive mastectomies rather than breast-conserving surgery [55,56,57], which requires daily postoperative radiation treatment for six weeks, and to be less likely to receive guideline-indicated curative-intent post-surgical radiation therapy [56,58]. Rural patients with endometrial [59] or prostate [60] cancer are similarly less likely to receive radiation than non-rural patients. Nearly three-quarters of this study’s cohort are in metro areas of population sizes 250,000 or greater. This supports the idea that patients in more urban areas are more likely to have access to radiation treatment. This presents an opportunity for improving access to treatment for NMSC via the increased adoption of IGSRT by rural dermatology practices.

Dermatology practices not set up for MMS could incorporate IGSRT to improve their community’s access to NMSC treatment. Similarly, dermatology clinics already offering MMS could serve additional patients by adding IGSRT to their practices, as IGSRT can be operated by a credentialed radiation therapist who has been appropriately trained in providing this level of care while dermatologists are concurrently treating other patients. Another consideration with respect to access is that most Mohs surgeons do not accept Medicare. Once IGSRT has consistently appropriate coverage by Medicare, it could help provide treatment to the Medicare population. Additionally, IGSRT offers short individual treatment sessions compared with MMS (15 min vs. 2–4 h, respectively) [15,61]. This technology also provides an alternative treatment option for patients unsuited or uninterested in surgical management. According to the American Society of Radiation Technologists Practice Standards, radiation therapists are healthcare professionals responsible for administering high doses of ionizing radiation to treat diseases, such as cancer, while working under the supervision of licensed practitioners [62]. In dermatology settings, radiation therapists can specialize in administering SRT using low-energy photons to treat NMSC. They follow DART Appropriate Use Criteria for the treatment of SCCs and BCCs [38,39]. The clinical expertise and technical skill set of radiation therapists, in collaboration with radiation oncologists, medical physicists, and the appropriate multi-disciplinary team, ensures the quality and safety of treatment.

One limitation of this study is its retrospective design. Unfortunately, due to the retrospective design, it was not possible to collect self-reported data about patient-level SES. Furthermore, only correlative conclusions can be drawn from a study of this design. Additionally, retrospective studies are at risk of bias. Some sources of potential bias include differing baseline characteristics between comparison groups, selection bias (if subjects are not representative of the population), missing chart data, recall bias (subjects misremembering information), data entry by different people, and loss of follow-up [63]. Therefore, future randomized controlled trials (RCTs) for IGSRT would offer valuable data or registry-based approaches with fewer sources of bias. Additionally, RCTs are not retrospective, so all desired data points can be planned for and appropriately collected, eliminating missing data, and patients can be questioned about their experience throughout the course of the study, minimizing recall bias.

## 5. Conclusions

This is the first large retrospective cohort study to evaluate the potential effects of neighborhood deprivation and patient comorbidity on freedom from recurrence following IGSRT treatment of NMSC. Overall, this study found that freedom from recurrence does not significantly vary by neighborhood socioeconomic status or comorbidity burden. In combination with previous cohort studies indicating the superiority of IGSRT, these findings suggest that IGSRT is an excellent first-line treatment option for patients who cannot or choose not to undergo surgical removal of NMSC regardless of their socioeconomic or comorbidities.

## Figures and Tables

**Figure 1 cancers-16-04037-f001:**
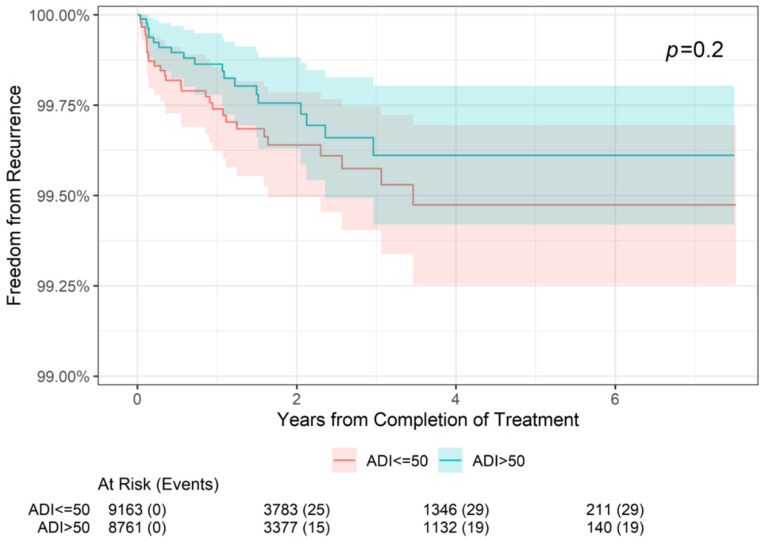
Freedom from recurrence over time of non-melanoma skin cancer treated with image-guided superficial radiation therapy by Area Deprivation Index (ADI) score. ADI ≤ 50 represents advantaged neighborhoods (high SES), and ADI > 50 represents disadvantaged neighborhoods. The “At Risk” value represents the sample size at the corresponding year of follow-up. The “Events” value represents the number of NMSC lesions that have recurred by the corresponding year of follow-up. The *p* value of 0.2 indicates that freedom from recurrence of the ADI > 50 group compared with the ADI ≤ 50 is not statistically significant.

**Figure 2 cancers-16-04037-f002:**
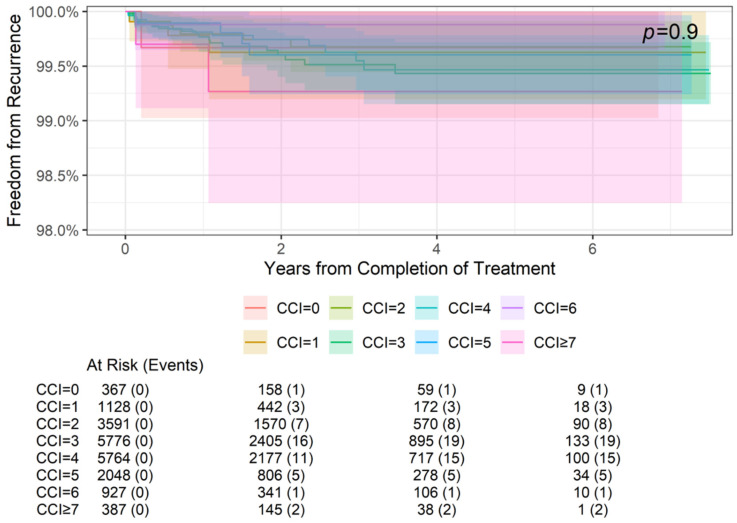
Freedom from recurrence over time of non-melanoma skin cancer treated with image-guided superficial radiation therapy by Charlson Comorbidity Index (CCI) score. Higher CCI scores represent higher comorbidity burdens. The “At Risk” value represents the sample size at the corresponding year of follow-up. The “Events” value represents the number of NMSC lesions that have recurred by the corresponding year of follow-up. The *p* value of 0.9 indicates that the differences in freedom from recurrence between CCI groups are not statistically significant.

**Figure 3 cancers-16-04037-f003:**
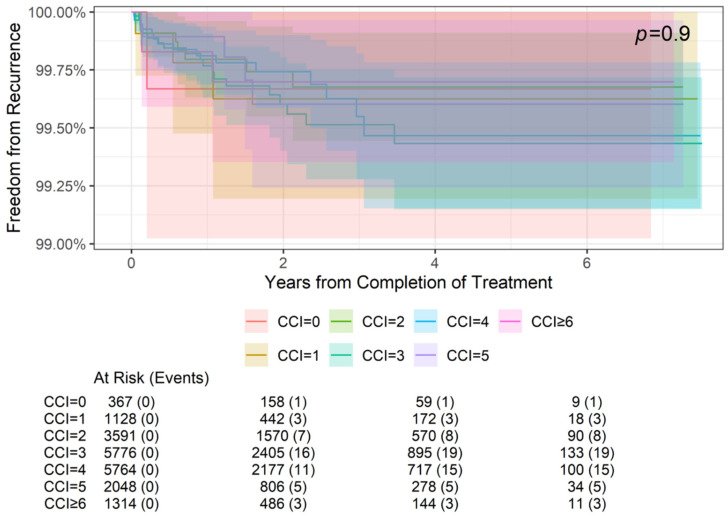
Freedom from recurrence over time of non-melanoma skin cancer treated with image-guided superficial radiation therapy by Charlson Comorbidity Index (CCI) scores 0–6+. Higher CCI scores represent higher comorbidity burdens. The “At Risk” value represents the sample size at the corresponding year of follow-up. The “Events” value represents the number of NMSC lesions that have recurred by the corresponding year of follow-up. The *p* value of 0.9 indicates that the differences in freedom from recurrence between CCI groups are not statistically significant.

**Table 1 cancers-16-04037-t001:** 2023 Rural–Urban Continuum Codes.

Code Number	In Metro Area	Adjacent to Metro Area	Population Size
1	Yes	N/A	≥1,000,000
2	Yes	N/A	250,000–1,000,000
3	Yes	N/A	<250,000
4	No	Yes	≥20,000
5	No	No	≥20,000
6	No	Yes	5000–19,999
7	No	No	5000–19,999
8	No	Yes	<5000
9	No	No	<5000

**Table 2 cancers-16-04037-t002:** Participant and disease characteristics.

Characteristic	Number of Lesions
Age, *n* (%)	
<65 Years	3158 (15.8)
≥65 Years	16,911 (84.2)
Tumor Location, *n* (%)	
Head/neck	12,787 (63.7)
Extremities	4142 (20.6)
Trunk	819 (4.1)
Stage, ^a^ *n* (%)	
0	
1	9885 (49.4)
2	5270 (26.4)
3	4635 (23.2)
Missing	198 (1.0)
Lesion Type, *n* (%)	
BCC	9885 (49.4)
SCC	5270 (26.4)
SCCIS	4635 (23.2)
2 ≥ NMSC types	198 (1.0)
Area Deprivation Index Score, median (IQR)	50.0 (24.0, 66.0)
Rural–Urban Continuum Codes, *n* (%)	
1	9712 (53.7)
2	3735 (20.7)
3	182 (1.0)
4	1080 (6.0)
5	248 (1.4)
6	1628 (9.0)
7	640 (3.5)
8	662 (3.7)
9	195 (1.1)
Unknown	1906 (9.54)
Charlson Comorbidity Index Score, *n* (%)	
0	367 (1.8)
1	1128 (5.6)
2	3591 (18.0)
3	5776 (28.9)
4	5764 (28.8)
5	2048 (10.2)
6	927 (4.6)
≥7	387 (1.9)

Abbreviations: BCC, basal cell carcinoma; SCC, squamous cell carcinoma; SCCIS, squamous cell carcinoma in situ; IQR, interquartile range. ^a^ Staging was based on the American Joint Committee on Cancer 8th edition non-Merkel Non-Melanoma Skin Cancer classification system.

**Table 3 cancers-16-04037-t003:** Most prevalent Charlson Comorbidity Index comorbidities in cohort by histologic type.

Comorbidity	Overall	BCC	SCC	SCCIS	Two or More NMSC Types
Diabetes	3563 (17.8%)	1537 (15.5%)	1002 (19.0%)	979 (21.1%)	45 (22.7%)
Solid Tumor Malignancy	1538 (7.7%)	752 (7.6%)	447 (8.5%)	329 (7.1%)	10 (5.1%)
Stroke or TIA	928 (4.6%)	432 (4.4%)	242 (4.6%)	237 (5.1%)	17 (8.6%)
Liver Disease	288 (1.4%)	118 (1.2%)	101 (1.9%)	67 (1.4%)	2 (1.0%)
Lymphoma	256 (1.3%)	103 (1.0%)	77 (1.5%)	73 (1.6%)	3 (1.5%)
Myocardial Infarction	214 (1.1%)	118 (1.2%)	60 (1.1%)	33 (0.7%)	3 (1.5%)
Leukemia	201 (1.0%)	69 (0.7%)	70 (1.3%)	60 (1.3%)	2 (1.0%)

Abbreviations: TIA, transient ischemic attack; AIDS, acquired immunodeficiency syndrome.

**Table 4 cancers-16-04037-t004:** Freedom from recurrence rates by socioeconomic status and comorbidities.

Index Score	2-Year Freedom from Recurrence	4-Year Freedom from Recurrence	6-Year Freedom from Recurrence
Area Deprivation Index Score, *n* (%)			
≤50 (*n* = 9163 at baseline)			
*n*, recurrence events	3783, 25	1346, 29	211, 29
% freedom from recurrence	99.37	98.00	87.26
>50 (*n* = 8761 at baseline)			
*n*, recurrence events	3377, 15	1132, 19	140, 19
% freedom from recurrence	99.56	98.33	86.43
Charlson Comorbidity Index (CCI) Score			
CCI 0 (*n* = 367 at baseline)			
*n*, recurrence events	158, 1	59, 1	9, 1
% freedom from recurrence	99.67	99.67	99.67
CCI 1 (*n* = 1128 at baseline)			
*n*, recurrence events	442, 3	172, 3	18, 3
% freedom from recurrence	99.63	99.63	99.63
CCI 2 (*n* = 3591 at baseline)			
*n*, recurrence events	1570, 7	570, 8	90, 8
% freedom from recurrence	99.74	99.68	99.68
CCI 3 (*n* = 5776 at baseline)			
n, recurrence events	2405, 16	895, 19	133, 19
% freedom from recurrence	99.60	99.43	99.43
CCI 4 (*n* = 5764 at baseline)			
*n*, recurrence events	2177, 11	717, 15	100, 15
% freedom from recurrence	99.74	99.47	99.47
CCI 5 (*n* = 2048 at baseline)			
*n*, recurrence events	806, 5	278, 5	34, 5
% freedom from recurrence	99.60	99.60	99.60
CCI 6 (*n* = 927 at baseline)			
*n*, recurrence events	341, 1	106, 1	10, 1
% freedom from recurrence	99.88	99.88	99.88
CCI ≥7 (*n* = 387 at baseline)			
*n*, recurrence events	145, 2	38, 2	1, 2
% freedom from recurrence	99.27	99.27	99.27

## Data Availability

The data presented in this study are available on request from the corresponding author.

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
