# Peer review of "The Impact of Socioeconomic Status and Comorbidities on Non-Melanoma Skin Cancer Recurrence After Image-Guided Superficial Radiation Therapy"

_cancers, 2024, doi:10.3390/cancers16234037_

Round 1

Reviewer 1 Report

Comments and Suggestions for Authors

The aim of this retrospective cohort study was to assess if there are relationships between patient comorbidities or socioeconomic status (SES) and outcomes from IGSRT treatment for their NMSCs.

Page 1 line 40: Charlson comorbidity index (CCI) instead of Charleston.

First consideration: CCI score is the results of age, comorbidities and their severity. Since the 84% of the analyzed population is ≥ 65- year older, I do not understand (Table 2) how 68.5 % of the subjects had score 0, the feeling is the authors considered only the number of comorbidities, while   from 50 to > 80-year-old   the score increases from 1 to 4.

This is a retrospective study on a large cohort population, some data could be unavailable, however the socioeconomic status and comorbidities impact, and moreover leaving alone impact more on the diagnosis delay, than on the efficacy of the treatment when adequately delivered.

I would not expect that subjects with a diagnosis in an early stage, and able to be treated with an effective treatment, like IGSRT is, should derived less benefit from the treatment than subjects leaving in less economic deprived neighborhoods, anyway it could be important to confirm this point.

Considering the comorbidities   reported and their score Table 2 and 3, I feel not confident in the reported scores, do all patients have uncomplicated diabetes? non metastatic solid tumors? How many had AIDS, Leukemia, or Lymphomas? Usually, this is the population with the higher incidence of NMSC, the frequencies reported in Table 3 seem not representative of the usual population seen in a dermato-oncology center.

Reference 12:  tumor control 99 %, median Follow-up 63 wks, median dose 5188 cGy , median fractions 20.

Table 2.  74.4 % of the subjects   treated lived in metro areas (1-2) and 68.5 % had CCSI of 0, this represents a favorable population.

Fig 2 unclear please redraw it in a clearer way.

The results of this study, confirm in a real world contest, the feasibility  and efficacy of  IGSRT, even  in  subjects living in  a  neighborhood-level deprivation,  to evaluate how  many patients were excluded and the reasons  remain an important issue to address.

Reviewer 2 Report

Comments and Suggestions for Authors

Dear authors,

I read with great interest your manuscript titled "The Impact of Socioeconomic Status and Comorbidities on Non-Melanoma Skin Cancer Recurrence After Image-Guided Superficial Radiation Therapy".

The study investigates the impact of socioeconomic status (SES) and comorbidities on non-melanoma skin cancer (NMSC) recurrence after image-guided superficial radiation therapy (IGSRT). 

The study is well designed and the results are clearly presented. A few points that are worth improving are the following:

1. The study focuses exclusively on freedom from recurrence. Broader outcomes, such as quality of life, functional outcomes, and treatment-related complications, are not considered, limiting the understanding of overall treatment efficacy. As this may not be feasible to be added in the current study, it should be commented in the discussion section.

2.Although the study highlights the importance of rural access to care, it does not fully explore any obstacles to implementing IGSRT in rural settings, such as financial constraints or provider availability. As this study is more of a public health access point-of-view, this should be commented upon.

3. The paper lacks a critical comparison to other treatment modalities (e.g., Mohs) in similar populations, which would enhance clinical applicability.

Reviewer 3 Report

Comments and Suggestions for Authors

1. Clarify the statistical results mentioned in the abstract. For instance, "no difference in freedom from recurrence in less vs. more deprived neighborhoods" could be specified with precise values.

2. In the introduction, the motivation is not clear and add more context on the current gaps in understanding SES impact on IGSRT outcomes, referencing specific prior studies.

3. Also addition of recent references that states why detecting NMSCs are also important for this kind of study:

Lin, Teng-Li, Chun-Te Lu, Riya Karmakar, Kalpana Nampalley, Arvind Mukundan, Yu-Ping Hsiao, Shang-Chin Hsieh, and Hsiang-Chen Wang. "Assessing the efficacy of the spectrum-aided vision enhancer (SAVE) to detect acral lentiginous melanoma, melanoma in situ, nodular melanoma, and superficial spreading melanoma." Diagnostics 14, no. 15 (2024): 1672.

Huang, Hung-Yi, Yu-Ping Hsiao, Riya Karmakar, Arvind Mukundan, Pramod Chaudhary, Shang-Chin Hsieh, and Hsiang-Chen Wang. "A Review of Recent Advances in Computer-Aided Detection Methods Using Hyperspectral Imaging Engineering to Detect Skin Cancer." Cancers 15, no. 23 (2023): 5634.

4. Provide further details in the "2.3 Data Collection" subsection about the algorithms used for data scraping.

4. Ensure all figures are clearly labeled, especially Figures 1-3 showing recurrence rates. Consider adding captions that explain the importance of each figure.

6. Expand on the retrospective design limitation by discussing potential biases introduced and how future studies might address them.

Reviewer 4 Report

Comments and Suggestions for Authors

1. The first conclusion of the paper, that SES did not impact outcomes for superficial RT for non-melanoma skin cancers, is both foregone and not really valid, because ALL patents were treated. This is therefore not a real examination for disparities. 

2. Table 3 lists several co-morbidities that were present at insignificant levels or completely absent. Shorten the table by removing co-morbidities present in less than 1% of patients, and simply listing what those were in the text.  

Round 2

Reviewer 3 Report

Comments and Suggestions for Authors

The authors have made all the changes suggested, hence the paper can be accepted.